# Enhanced Stability and Compatibility of Montelukast and Levocetirizine in a Fixed-Dose Combination Monolayer Tablet

**DOI:** 10.3390/pharmaceutics16070963

**Published:** 2024-07-21

**Authors:** Tae Han Yun, Moon Jung Kim, Jung Gyun Lee, Kyu Ho Bang, Kyeong Soo Kim

**Affiliations:** Department of Pharmaceutical Engineering, Gyeongsang National University, 33 Dongjin-ro, Jinju 52725, Republic of Korea; xogks7702@naver.com (T.H.Y.); kmjksk137@naver.com (M.J.K.); leepipi87@naver.com (J.G.L.); khbang0095@gnu.ac.kr (K.H.B.)

**Keywords:** montelukast, levocetirizine, fixed-dose combination, stability, compatibility

## Abstract

The purpose of this study was to enhance the stability of montelukast and levocetirizine for the development of a fixed-dose combination (FDC) monolayer tablet. To evaluate the compatibility of montelukast and levocetirizine, a mixture of the two drugs was prepared, and changes in the appearance characteristics and impurity content were observed in a dry oven at 60 °C. Excipients that contributed minimally to impurity increases were selected to minimize drug interactions. Mannitol, microcrystalline cellulose, croscarmellose sodium, hypromellose, and sodium citrate were chosen as excipients, and montelukast–levocetirizine FDC monolayer tablets were prepared by wet granulating the two drugs separately. A separate granulation of montelukast and levocetirizine, along with the addition of sodium citrate as a pH stabilizer, minimized the changes in tablet appearance and impurity levels. The prepared tablets demonstrated release profiles equivalent to those of commercial products in comparative dissolution tests. Subsequent stability testing at 40 ± 2 °C and 75 ± 5% RH for 6 months confirmed that the drug content, dissolution rate, and impurity content met the specified acceptance criteria. In conclusion, the montelukast–levocetirizine FDC monolayer tablet developed in this study offers a potential alternative to commercial products.

## 1. Introduction

In recent decades, the prevalence of respiratory and allergic diseases, such as allergic rhinitis and asthma, has increased [1]. While not directly life-threatening, respiratory allergic diseases significantly deteriorate quality of life [2]. Two prominent types of these allergic diseases are asthma and rhinitis [3]. Asthma, a common chronic condition, induces persistent difficulty in breathing, coughing, and other respiratory symptoms [4]. Allergic rhinitis, which is characterized by inflammation of the nasal mucosa, results in symptoms such as nasal discharge and sneezing [5]. Although asthma and allergic rhinitis can occur independently, up to 40% of rhinitis patients may also have asthma, and as much as 80% of asthma patients may experience rhinitis, which indicates a high co-occurrence within the same patient population [6]. Therefore, integrated treatment of allergic rhinitis and asthma is recommended according to the Allergic Rhinitis and its Impact on Asthma (ARIA) guidelines, which advise combining antihistamines and leukotriene receptor antagonists [7,8]. Levocetirizine, a second-generation antihistamine, is frequently used to treat allergic rhinitis [9], while montelukast, a CysLT1 receptor antagonist, helps improve asthma symptoms by blocking the binding of leukotrienes to a receptor [10]. In patients with persistent allergic rhinitis, combination therapy with montelukast and levocetirizine is more effective than montelukast monotherapy [11]. Additionally, fixed-dose combination (FDC) therapy with montelukast and levocetirizine has proven its superiority through phase III clinical trials for treating allergic rhinitis in asthma patients [12,13]. Furthermore, in terms of safety, there is no difference between short-term administration (1–4 weeks) and long-term administration (3–6 months), thus making it an effective and safe drug for improving the symptoms of perennial allergic rhinitis with asthma, even after long-term use [14].

The purpose of this study was to investigate the compatibility and stability of montelukast and levocetirizine for the development of an FDC monolayer tablet to treat allergic rhinitis and asthma. Previous studies have investigated the FDC bilayer tablets of montelukast and levocetirizine, but research on monolayer tablets has not yet been sufficiently conducted [15,16]. While a combination monolayer tablet offers economic advantages when compared to taking each commercial product separately, challenges such as decreased stability may emerge due to the interactions between the two main drugs [17]. Therefore, in the initial study stage, an impurity test was performed to confirm the compatibility of the two drugs and select excipients to ensure the stability of the combination tablet [18]. Afterward, granules were prepared with selected excipients through wet granulation, which is a common granulation method, and, subsequently, the tablets were compressed [19]. The stability of the prepared tablets was confirmed by evaluating the changes in appearance characteristics and impurity content under stress conditions at 60 °C. Comparative dissolution tests were conducted on the final selected formulation and the commercial product under the pH conditions of the entire gastrointestinal tract to predict the bioequivalence. Subsequently, the tablets were stored at 40 ± 2 °C and 75 ± 5% RH for 6 months to assess any decrease in the drug content, reduction in the dissolution rate, and increase in the impurity content.

## 2. Materials and Methods

### 2.1. Materials

Montelukast sodium was purchased from Macleods Pharmaceuticals Ltd. (Mumbai, India), and levocetirizine dihydrochloride was obtained from RA Chem Pharma Ltd. (Polepally, Telangana, India). The impurity standards of montelukast for the HPLC analysis of the impurity were purchased from SynZeal Research Private Ltd. (Ahmedabad, Gujarat, India). The excipients used in the formulation study, including microcrystalline cellulose (abbreviated as Avicel^®^), lactose, hypromellose (HPMC 2910 P645), povidone, croscarmellose sodium, sodium starch glycolate, crospovidone, and magnesium stearate, were kindly provided by Hanmi Pharmaceutical Co., Ltd. (Hwaseong, Republic of Korea). Mannitol (Mannitol 100SD) and hydroxypropyl cellulose were supplied by Cosmax Pharma Co., Ltd. (Cheongju, Republic of Korea). All pH stabilizers were purchased from Daejung Chemicals & Metals Co., Ltd. (Siheung, Republic of Korea). Acetonitrile and methanol were sourced from Thermo Fisher Scientific (Waltham, MA, USA), and the trifluoroacetic acid was obtained from Daejung Chemicals & Metals Co., Ltd. (Siheung, Republic of Korea). The deionized water used in the laboratory was produced using a distillation device. All other chemicals were of analytical grade.

### 2.2. Compatibility Study between Montelukast and Levocetirizine

A total of 1 g of montelukast, 1 g of levocetirizine, and 1 g of their 1:1 mixture was individually placed in high-density polyethylene (HDPE) containers. These samples were then stored in a 60 °C dry oven (HB-502M; Hanbaek Scientific Technology, Bucheon, Republic of Korea). The HDPE containers, with the samples, remained unopened throughout the storage period. Afterward, these samples were collected at each time point (1–4 weeks) to visually observe the appearance characteristics and to assess the impurity content. Details on the method for assessing the impurity content are described in Section 2.8, ‘Impurity Test’.

### 2.3. Compatibility Study of Excipients

Montelukast and levocetirizine were mixed with microcrystalline cellulose, mannitol, lactose, hydroxypropyl cellulose, hypromellose, povidone, croscarmellose sodium, sodium starch glycolate, crospovidone, citric acid, meglumine, and sodium citrate in a 2:1:20 (*w*/*w*/*w*) ratio, and this solution was then placed into individual glass vials. The vials were sealed with a plastic cap. The vials containing the samples were stored in a 60 °C dry oven and remained unopened. These samples were collected at each time point (1–4 weeks) according to the method described in Section 2.7 ‘Impurity Test’ in order to assess the impurity content.

### 2.4. Preparation of Montelukast–Levocetirizine FDC Monolayer Tablets

Through compatibility testing, excipients were selected that could minimize the increase in impurity content of montelukast and levocetirizine. Using the selected excipients, granules of montelukast and levocetirizine were prepared through wet granulation. Montelukast granules were created using a high shear mixer (TOP-05HSM; Hankook P.M ENG CO., Gunpo, Republic of Korea). Ingredients including montelukast, Mannitol 100SD, Avicel PH101, croscarmellose sodium, and sodium citrate were blended, with the impeller and chopper speeds set at 230 rpm and 1500 rpm, respectively. Following this, a solution of HPMC 2910 P645 that had been dissolved in water was added to the mixture and kneaded. The resulting wet granules were then dried in a dry oven at 60 °C until the water content was reduced to below 1.0%. To set the drying temperature, 60 °C was chosen as the optimal temperature (based on the findings from the preliminary study stage within the temperature range of 50–70 °C) at which the water content would drop below 1.0% within 1 h. Subsequently, the dried granules were sieved through a 16-mesh screen and stored in polyethylene bags containing silica gel. The preparation of levocetirizine granules followed a similar procedure as that of the montelukast granules. Levocetirizine granules were also prepared using a high-shear mixer. Levocetirizine, Mannitol 100SD, Avicel PH101, and meglumine were blended, with the impeller and chopper speeds set at 120 rpm and 1500 rpm, respectively. The subsequent processes were identical to those used in the preparation of the montelukast granules. Once the granules were prepared, the montelukast and levocetirizine granules were first combined in a blender (AR403; ERWEKA, Langen, Germany). Magnesium stearate was then added for the final mixing. Afterward, the final mixed granules were compressed into tablets using a rotary tablet press machine (ZP10; M.D Korea Co., Hwaseong, Republic of Korea). The compression speed of the rotary tablet press machine was set at 25 rpm, and the tablet hardness was adjusted to 8–11 kilopond.

### 2.5. Stability Testing of the Prepared Montelukast–Levocetirizine FDC Monolayer Tablets

The prepared montelukast–levocetirizine FDC monolayer tablets were placed in HDPE containers and sealed with a plastic cap. Subsequently, they were stored in a drying oven at 60 °C for 4 weeks. At 1 week, 2 weeks, and 4 weeks, the tablets were collected to evaluate the changes in appearance characteristics and impurity content. The most stable tablet was then selected as the final formulation.

### 2.6. SEM-EDS Mapping of the Tablet

The surface morphology and elemental composition of the samples were analyzed using scanning electron microscopy (SEM) combined with energy dispersive spectroscopy (EDS). The analyses were conducted using a Tescan-MIRA3 SEM (TESCAN KOREA, Seoul, Republic of Korea). Initially, the montelukast–levocetirizine FDC monolayer tablets were sectioned to allow observations of the cross-section. Subsequently, double-sided adhesive tape was utilized to fix the samples. To render the samples electrically conductive, a platinum coating (4 min at 25 mA) was applied using a sputter coater (K575X; EmiTech, Madrid, Spain) at a speed of 6 nm/min under vacuum (7 × 10^−3^ mbar) conditions.

### 2.7. Drug Content Uniformity Test

The content uniformity test solution was prepared by dissolving one tablet in a 100 mL volumetric flask with a diluent consisting of a 3:1 mixture of methanol and water. The tablet should be placed in a volumetric flask in its original, unground state, and it should be shaken until it is fully disintegrated in the diluent and the shape of the tablet is no longer visible. The test solution was prepared using 10 randomly selected tablets, resulting in a total of 10 test solutions. Each test solution was filtered through a nylon membrane filter and then analyzed for drug content, using the HPLC method, to evaluate the content uniformity of each drug. The wavelength used for the drug content analysis was selected as 225 nm, considering the UV spectra of montelukast and levocetirizine (Appendix A). Based on the measured content, the acceptance value (AV) was evaluated according to the following formula:$$AV=\left|M-\bar{x}\right|+k\cdot s$$
where M is the reference value (usually the target content), $\bar{x}$ is the mean content of the sample units, k is the acceptability constant (2.4 for 10 units), and s is the standard deviation of the sample units.

### 2.8. Impurity Test

The compatibility study test solutions were prepared by dissolving each of the mixtures that had been prepared in Section 2.2 and Section 2.3 into diluents of each drug in order to achieve a drug concentration of 1000 μg/mL. Montelukast was diluted in a 9:1 mixture of methanol and water, while levocetirizine was diluted in a 3:7 mixture of acetonitrile and water. The stability study test solutions for the monolayer tablet were prepared by grinding the tablet into a powder and dissolving it in a diluent of each drug to achieve a drug concentration of 1000 μg/mL for each drug. To evaluate the impurities of montelukast and levocetirizine, different HPLC conditions must be used for each. Therefore, separate test solutions were prepared for each drug for the purposes of the compatibility test and stability test. Hence, different diluents were also used for each drug. Standard solutions for impurity content calculation were prepared by dissolving montelukast and levocetirizine drug powders in their respective diluents to achieve a drug concentration of 1 μg/mL (0.1% of the test solution). Each test solution was then filtered through a nylon membrane filter and analyzed via HPLC to assess the impurities in montelukast and levocetirizine (Table 1) [20,21].

### 2.9. The Dissolution Test

The dissolution of the montelukast–levocetirizine FDC monolayer tablets was performed using a USP dissolution apparatus II (RCZ-6N; Pharmao Industries Co., Liaoyang, China). For the montelukast dissolution test, however, media with a pH of 1.2, 4.0, and 6.8, as well as 0.5% SLS (5 g of sodium lauryl sulfate dissolved in 1 L water), were used. For the levocetirizine dissolution test, media with a pH of 1.2, 4.0, and 6.8, as well as water, were used. All of the dissolution media used were 900 mL in volume. The pH 1.2 solution was prepared using 0.1 M hydrochloric acid and sodium chloride, while the pH 4.0 solution utilized a 0.05 M sodium acetate buffer solution and the pH 6.8 solution was made by combining a 0.2 M potassium dihydrogen phosphate solution with a 0.2 M sodium hydroxide solution. The pHs of 1.2, 4.0, and 6.8 represent the gastrointestinal tract environment [22]. The temperature of the dissolution media was adjusted to 37 ± 0.5 °C. In the stability test, only 0.5% SLS (for montelukast) and water (for levocetirizine) were used [23]. The montelukast–levocetirizine FDC monolayer tablets were placed in the dissolution media at a paddle speed of 50 rpm. In the comparative dissolution test, Singulair^®^ tablets were used as the reference commercial product for montelukast, and Xyzal^®^ tablets were used for levocetirizine. At predetermined times, 3 mL of the medium was sampled, filtered through a nylon membrane filter, and analyzed via HPLC to assess the drug content in montelukast and levocetirizine (Table 1).

### 2.10. Stability Testing

Accelerated stability studies were conducted on the montelukast–levocetirizine FDC monolayer tablet in accordance with ICH guidelines [24,25]. The tablets were packaged in HDPE containers with silica gel and stored at 40 ± 2 °C and 75 ± 5% RH for 6 months. Subsequently, the samples were taken out at predetermined time points (initial, 2 months, 4 months, and 6 months) to confirm the drug content, dissolution rate, and impurity content.

## 3. Results

### 3.1. Compatibility Study between Montelukast and Levocetirizine

Compatibility studies that examine the interactions between two drugs in a combination formulation are crucial during the preliminary stages of formulation development [26]. Before selecting excipients, the interactions between the drugs were examined by observing the changes in appearance characteristics and the impurity content while storing the individual drugs and their mixture at 60 °C (Figure 1).

Montelukast exhibited a yellowish color and a melting point of 60 °C, whereas levocetirizine showed no significant changes in its appearance characteristics. However, the mixture of montelukast and levocetirizine displayed color changes at all examined time points. The color of the mixture was confirmed to be a darker yellow compared to that of montelukast alone. This was due to an accelerated color change resulting from the interaction between montelukast and levocetirizine. Due to anticipating a potential increase in the impurity levels because of this phenomenon, the total impurity content in both the drugs was examined. Before analyzing the impurity content, the impurity standards specified in the United States Pharmacopeia (USP) for each drug were analyzed via HPLC to check the retention times and to ensure well-separated results, with no overlap between the impurity peaks and the drug peaks (Figure 2). The impurities of montelukast that are specified in the montelukast sodium tablet section of the USP are sulfoxide impurities, montelukast ketone impurities, and cis-isomer impurities. The impurities of levocetirizine specified in the levocetirizine dihydrochloride tablet section of the USP were not defined. Refer to Table 1 for the HPLC methods for the two drugs, and also note that the HPLC conditions for the impurities of montelukast and levocetirizine were different.

For montelukast, the total impurity content, as an individual drug, increased marginally from 0.44% to 0.53%, i.e., by approximately 0.1% (Figure 3). On the other hand, the total impurity content in the mixture appeared to have doubled when compared to the initial content, rising from 0.41% to 0.80%. For levocetirizine, a similar pattern of results was observed as had been with montelukast. For the individual drugs, there was a slight increase from 0.18% to 0.21%, but there was also a significant increase of approximately 0.4% for the mixture, i.e., from 0.16% to 0.56%. When the two drugs were mixed, the increase in impurity content indicated that the interaction between them may decrease their stability. Additionally, changes in the appearance characteristics of a drug are crucial factors in terms of the drug product quality [27]. However, when mixing two drugs, the changes in appearance characteristics becomes severe, which is detrimental to the quality of the drug. Considering these results, it was found that it is essential to select excipients that do not decrease the stability due to the influence of potential interactions when combining montelukast and levocetirizine [28].

### 3.2. Compatibility Study of Excipients

The results of the compatibility study between montelukast and levocetirizine confirmed a potential risk to stability due to the interaction between the two drugs. Therefore, it was crucial to select excipients that enhance stability. A mixture of montelukast, levocetirizine, and excipients in a ratio of 2:1:20 was stored at 60 °C for 4 weeks, and the impurity content was assessed in the samples. In the compatibility study on the excipients, a ternary mixture of montelukast, levocetirizine, and excipients was analyzed. Impurity analysis results were compared for each excipient (Figure 4).

The total impurity content in montelukast and levocetirizine in the filler group (microcrystalline cellulose, mannitol, and lactose) was low, at 0.4% to 0.6% and 0.2% to 0.5%, respectively, thus showing a lower impurity content than the binder and disintegrant groups. Additionally, it was confirmed that the impurity content did not increase significantly over time. Therefore, it was also confirmed that the impurity content would not significantly increase, regardless of the type used. Hypromellose was selected as a binder because it minimizes increases in the impurity content of montelukast and levocetirizine. In particular, povidone exhibited a significant increase in the total impurity content when compared to the other binders. This occurred due to the hydrogen peroxide present in povidone promoting the oxidative degradation of the easily oxidizable parts of montelukast’s structure, thus leading to a substantial increase in impurity content [29,30,31,32]. These results also affected the total impurities in levocetirizine, which significantly increased to approximately 4.0% when compared to the other binders. Croscarmellose sodium was selected as the disintegrant because it resulted in the smallest impurity increase for both montelukast and levocetirizine. Similarly, as copovidone also belongs to the povidone derivative, it also led to an increase in the amount of impurities. In addition to fillers, binders, and disintegrants, screening was also conducted for acidifying and alkalinizing the agents that serve as pH stabilizers. This was performed because montelukast is stable in alkaline conditions and levocetirizine is stable in acidic conditions, which means that pH stabilizers can help enhance overall stability [33]. As a result, sodium citrate was selected as an alkalizing agent due to it causing a minimal increase in the impurity content, and meglumine was selected as an acidifying agent. Overall, it was confirmed that excipients can have some effect on increasing the impurity content in montelukast and levocetirizine. Among them, microcrystalline cellulose, mannitol, hypromellose, croscarmellose sodium, sodium citrate, and meglumine, which minimally increases the impurity content, were selected as excipients.

### 3.3. Montelukast–Levocetirizine FDC Monolayer Tablets

Through excipient screening, the excipients that minimally increased the impurity content were identified, thereby leading to the formulation of four types of montelukast–levocetirizine FDC monolayer tablets (Table 2). “q.s” listed in the table means “quantity sufficient”, and the hyphen (-) indicates that it was not added.

A compatibility study of montelukast and levocetirizine confirmed that, when mixed, the two drugs interact, resulting in a decrease in stability. Consequently, to prevent interaction and ensure stability, specific features were assigned to each formulation for stability comparison. In order to confirm the interactions between montelukast and levocetirizine, four formulations were prepared: one with post-blended levocetirizine raw material, one with granulated levocetirizine, one with sodium citrate (an alkalizing agent) being added to montelukast granules, and one with meglumine (an acidifying agent) being added to levocetirizine granules. To confirm the stability of the four formulations, the tablets were stored in a dry oven at 60 °C for 4 weeks. Afterward, the impurity and appearance characteristics that could indicate a stability decrease due to interactions between the two drugs were compared (Figure 5 and Figure 6). After comparing stability, the final formulation with the highest stability was selected based on the results from each formulation.

As a result of comparing the impurity content of the four formulations, the total impurity content in montelukast and levocetirizine in F1 was observed to have significantly increased when compared to the initial measurement, unlike the remaining tablets. For montelukast, the total impurity content increased from 0.51% to 3.23%; for levocetirizine, it increased significantly from 0.53% to 1.98%. Unlike the other tablets, F1 was prepared by post-blending the levocetirizine raw material rather than granulating it. Therefore, the physical distance between montelukast and levocetirizine within the tablet was reduced, resulting in a significant interaction. Unlike F1, F2 had the same excipients but differed in its preparation method. F2 included granulated levocetirizine to enhance the physical separation from montelukast, thereby diminishing the interactions between them. The difference in preparation method affected the impurity results, resulting in a lower impurity content when compared to F1. Next, the impurity content of F3, in which the pH stabilizer sodium citrate (an alkalinizing agent) was added to montelukast granules, and F4, in which the pH stabilizer meglumine (an acidifying agent) was added to levocetirizine granules, were compared. In the case of F3, the total impurity content i montelukast did not significantly increase from 0.21% to 1.22% when compared to the initial measurement. This is because the sodium citrate contributed to the stability of montelukast. On the other hand, F4, which contained meglumine, showed a higher impurity content than F3. These results show that meglumine is less helpful than sodium citrate for the stability of montelukast. The total impurity content of levocetirizine in F3 was slightly higher than in F4. Due to the fact that meglumine contributed to the stability of levocetirizine, F4 exhibited a lower impurity content. However, at 4 weeks, the total impurity content in levocetirizine was 0.84% for F3 and 0.68% for F4, which does not represent a significant difference. Consequently, sodium citrate was identified as an effective pH stabilizer, and it was used to facilitate an enhanced stability in the interactions between the two drugs. Furthermore, to maintain physical separation between the two drugs within the tablet, it was determined that granulating the drugs independently contributed to improved stability. In addition to comparing the impurity content, changes in the appearance characteristics of the tablets were also examined.

A comparison of the appearance characteristics of Tablets F3 and F4 showed that F3 maintained its appearance characteristics without any changes, both initially and after being stored at 60 °C for 4 weeks, and it overall exhibited a white appearance. On the other hand, the F4 tablet showed a faint yellow color initially, and the color change to yellow accelerated over 4 weeks. The mechanical properties of F3 and F4 tablets are included in Appendix A. As can be seen from the results when comparing the appearance characteristics, it was found that the addition of sodium citrate to the montelukast granules was helpful in preventing changes in appearance characteristics. Considering the overall results, preventing interactions through the separate granulation of the two drugs and the addition of sodium citrate contributed to the stability of both montelukast and levocetirizine, thus leading to the development of a stable monolayer tablet.

### 3.4. Distribution of Montelukast and Levocetirizine inside the Tablet

SEM-EDS mapping was used to visually confirm whether montelukast and levocetirizine were evenly distributed inside the tablet (Figure 7). The samples shown below are cross-sectional measurements of the F3 tablet, i.e., the final formulation with the highest stability.

The structure of montelukast sodium is C_35_H_36_ClNO_3_S·Na, and the structure of levocetirizine dihydrochloride is C_21_H_25_ClN_2_O_3_·2HCl. Therefore, the characteristic elements, i.e., the S (sulfur) of montelukast and the Cl (chlorine) of levocetirizine dihydrochloride, were analyzed through a mapping method. S (sulfur) is represented in yellow, and Cl (chlorine) is shown in green. It was confirmed that both elements were evenly distributed and well dispersed throughout the cross-section of the tablet, and these results also show that the two drugs were evenly mixed inside the tablet. After visually confirming the distribution of the two drugs using EDS, the uniformity of the content of montelukast and levocetirizine was confirmed through additional tests. As a result, the acceptance values of montelukast and levocetirizine were 6.1 and 7.4, respectively; thus, they did not exceed 15, which confirms an appropriate uniformity.

### 3.5. Comparative Dissolution Study

In vitro drug release tests were conducted to compare and confirm the dissolution patterns and the similarity of dissolution rates between Singulair tablets (10 mg of montelukast in a commercial tablet), Xyzal tablets (5 mg of levocetirizine in a commercial tablet), and the F3 tablet (Figure 8 and Figure 9). Study on the release rate of montelukast showed that both the Singulair tablet and the F3 tablet had very low dissolution rates in solutions that had a pH of 1.2, 4.0, and 6.8. The low dissolution rate of the Singulair tablets in a pH buffer solution was similar to other results in the literature [34]. The reason for this might be attributed to the low solubility of montelukast in physiological pH solutions that represent the gastrointestinal tract. Due to this characteristic of montelukast, to ensure bioequivalence with the Singulair tablet, a similar drug release profile must be demonstrated in 0.5% SLS. This should be sufficient as many other studies have also only conducted dissolution tests in a 0.5% SLS solution to ensure bioequivalence [35,36]. In the 0.5% SLS solution (the standard dissolution medium specified in the USP), both the Singulair tablet and the F3 formulation achieved a drug release rate of over 85% within 15 min, thus ensuring a dissolution equivalence between the two products [35]. Study on the release rate of levocetirizine showed that nearly a 100% dissolution was achieved within 15 min in all solutions, thus indicating dissolution equivalence as the dissolution exceeded 85% within 15 min. These results demonstrate that the F3 tablet exhibits a release rate that is particularly similar to that of commercial products, which suggests equivalence in their dissolution profiles.

### 3.6. Stability Study

In accordance with ICH guidelines, the F3 tablets, which demonstrated an excellent stability and dissolution equivalent to commercial products, were subjected to a stability test for 6 months under conditions of 40 ± 2 °C and 75 ± 5% RH. This test evaluated the drug content, dissolution rate, and impurity content (Figure 10) [37]. The criteria for drug content, dissolution rate, and impurity content were based on the United States Pharmacopeia (USP) for each drug.

The acceptance criteria for montelukast and levocetirizine drug content were set at 92.5–107.5% and 90.0–110.0%, respectively. Additionally, the acceptance criteria for the dissolution rate of montelukast and levocetirizine were set at 85% or more within 20 min and 85% or more within 30 min, respectively. For montelukast, the impurity criteria required sulfoxide to be below 2.0% and the total impurity content to be below 3.0%. For levocetirizine, an unspecified impurity was required to be below 0.3%, and the total impurity content had to be below 1.0%. It was confirmed that the drug content of montelukast and levocetirizine remained within the acceptance criteria at all time points, without significant decreases. The dissolution test results indicated that the dissolution rate of levocetirizine did not significantly decrease and remained within the acceptance criteria. On the other hand, the dissolution rate of montelukast decreased over time because the amorphous drug was converted to a crystalline form due to exposure to a high humidity environment [38]. However, after 6 months, the dissolution rate of the montelukast was 88%, which did not exceed the acceptance criteria. The montelukast impurity test results up to the 6-month time point showed that the sulfoxide impurity and total impurity content was 1.23% and 1.61%, respectively, which means both were within the acceptance criteria. Similarly, the levocetirizine impurity test results showed that the highest unspecified impurity and total impurity content was 0.25% and 0.61%, respectively, which means they were stable and within the acceptance criteria for up to 6 months. Consequently, the stability test results for the F3 tablet show that, over a period of 6 months at 40 ± 2 °C and 75 ± 5% RH, the drug content, dissolution rate, and impurity content all remain within acceptable limits, thus confirming that the stability does not significantly decrease.

## 4. Conclusions

The purpose of this study was to improve the stability of montelukast and levocetirizine in the formulation of an FDC monolayer tablet. To enhance the stability of these two drugs, it is essential to minimize the interactions that occur when the drugs come into contact with each other. It was confirmed that contact between the two drugs leads to changes in appearance characteristics and an increase in the impurity content. Therefore, through impurity testing, excipients that minimally increase the impurity content were selected to ensure that stability is maintained. In addition, montelukast and levocetirizine were granulated separately and sodium citrate was added as a pH stabilizer to minimize the increase in the impurity content and to prevent any change in appearance characteristics. The stability (in terms of drug content, dissolution rate, and impurity content) of the prepared montelukast–levocetirizine FDC monolayer tablet did not significantly decrease under conditions of 40 ± 2 °C and 75 ± 5% RH. Additionally, comparative dissolution tests on commercial products confirmed similar release profiles. Based on these results, it can be expected that commercial products and the montelukast–levocetirizine FDC monolayer tablets are bioequivalent. Consequently, the montelukast–levocetirizine FDC monolayer tablet, which was developed through a simple wet granulation process, offers a viable option for market introduction with assured stability.

## Figures and Tables

**Figure 1 pharmaceutics-16-00963-f001:**
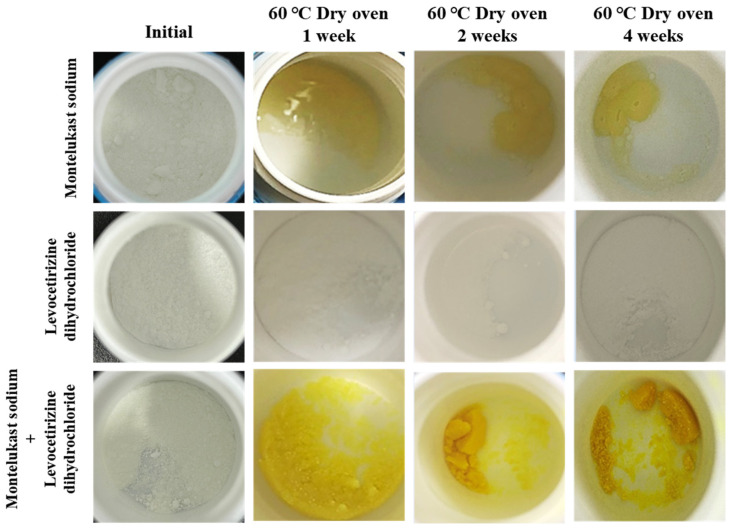
Characteristics of the changes in the appearance of montelukast alone, levocetirizine alone, and both in a 1:1 mixture.

**Figure 2 pharmaceutics-16-00963-f002:**
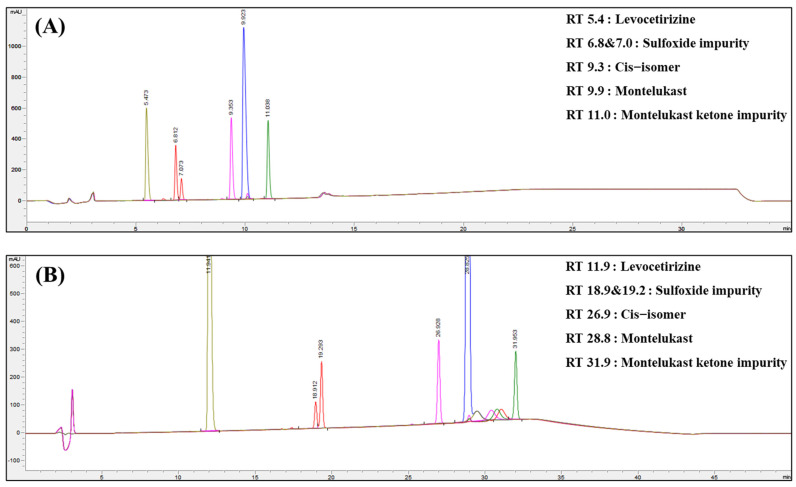
(**A**) HPLC chart for the montelukast impurity analysis, and (**B**) HPLC chart for the levocetirizine impurity analysis.

**Figure 3 pharmaceutics-16-00963-f003:**
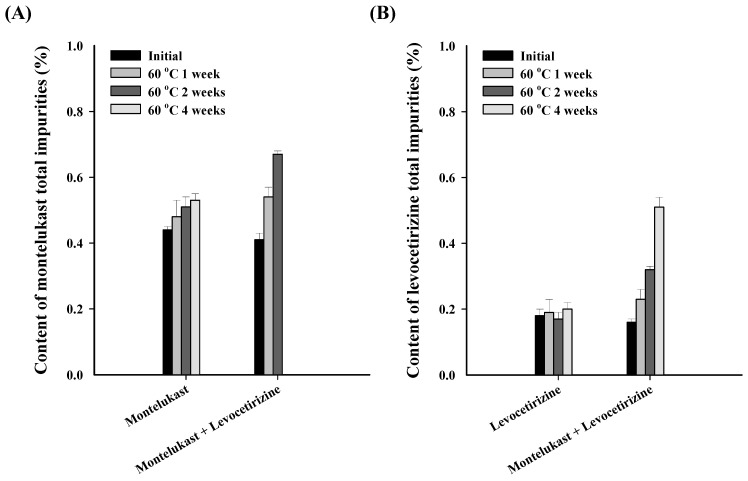
Results of (**A**) the total impurity content in montelukast, and (**B**) the total impurity content in levocetirizine for each drug and their mixture.

**Figure 4 pharmaceutics-16-00963-f004:**
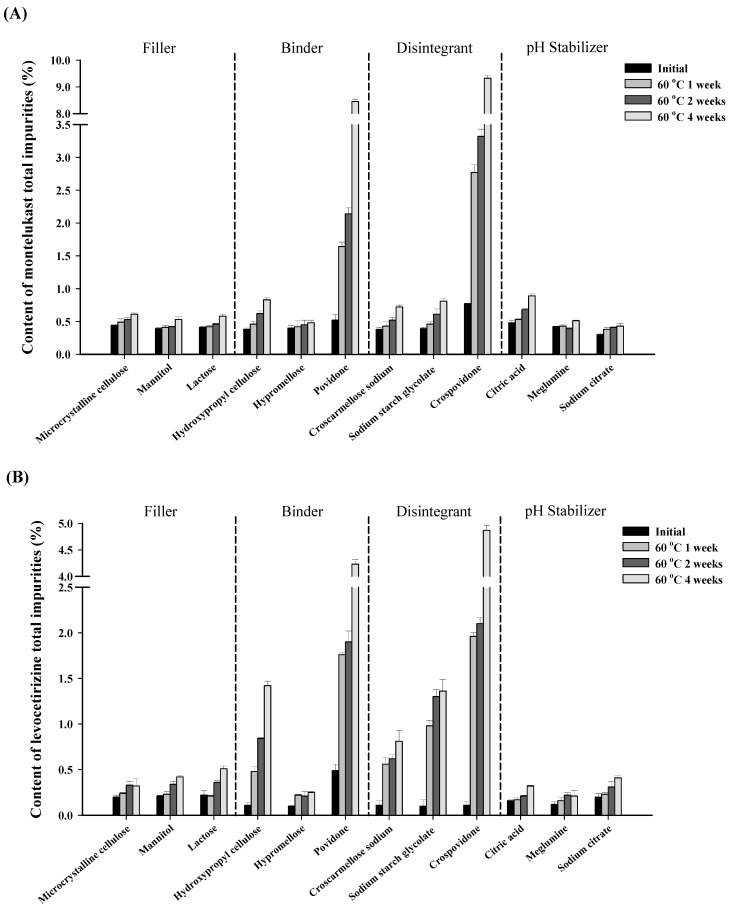
Results of the (**A**) total impurity content in montelukast, and (**B**) the total impurity content in levocetirizine for each excipient.

**Figure 5 pharmaceutics-16-00963-f005:**
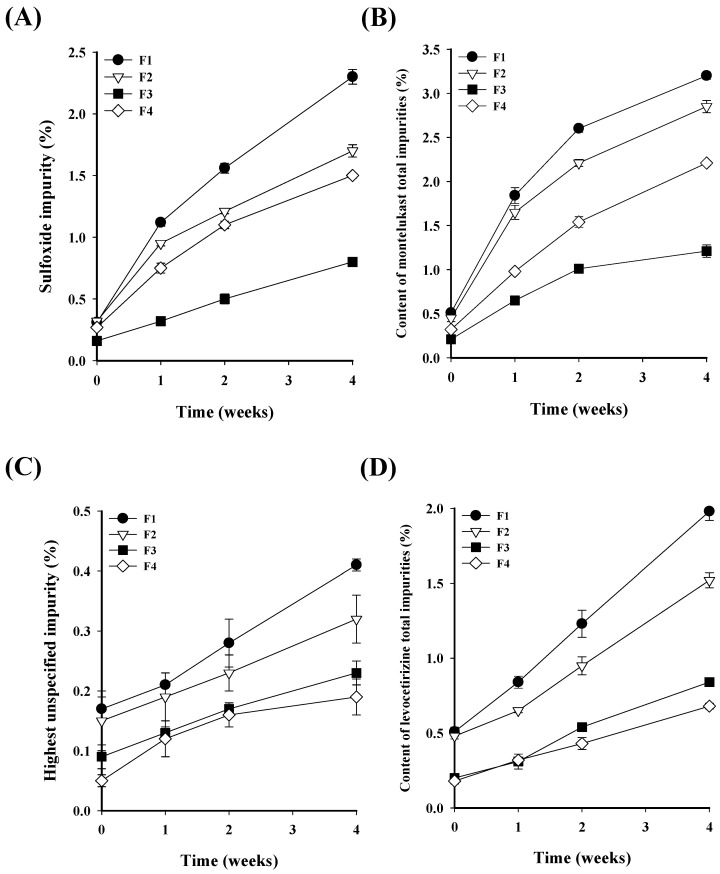
Comparison of (**A**) montelukast sulfoxide impurity, (**B**) the total impurity content of montelukast, (**C**) the specific impurity content of levocetirizine, and (**D**) the total impurity content of levocetirizine in the four formulations.

**Figure 6 pharmaceutics-16-00963-f006:**
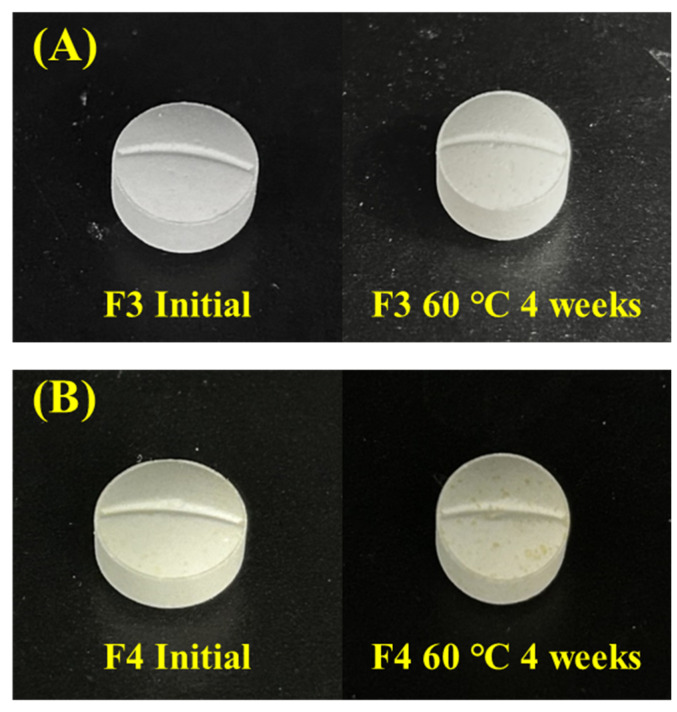
Comparison of the changes in appearance characteristics between (**A**) the F3 tablet and (**B**) the F4 tablet.

**Figure 7 pharmaceutics-16-00963-f007:**
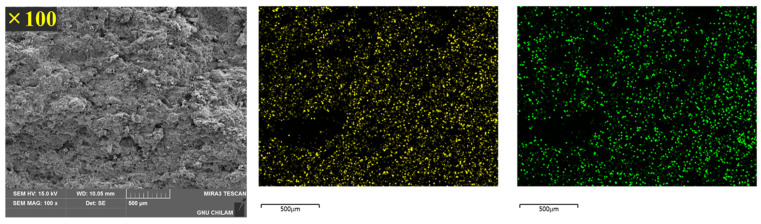
SEM-EDS mapping image of a F3 tablet cross-section.

**Figure 8 pharmaceutics-16-00963-f008:**
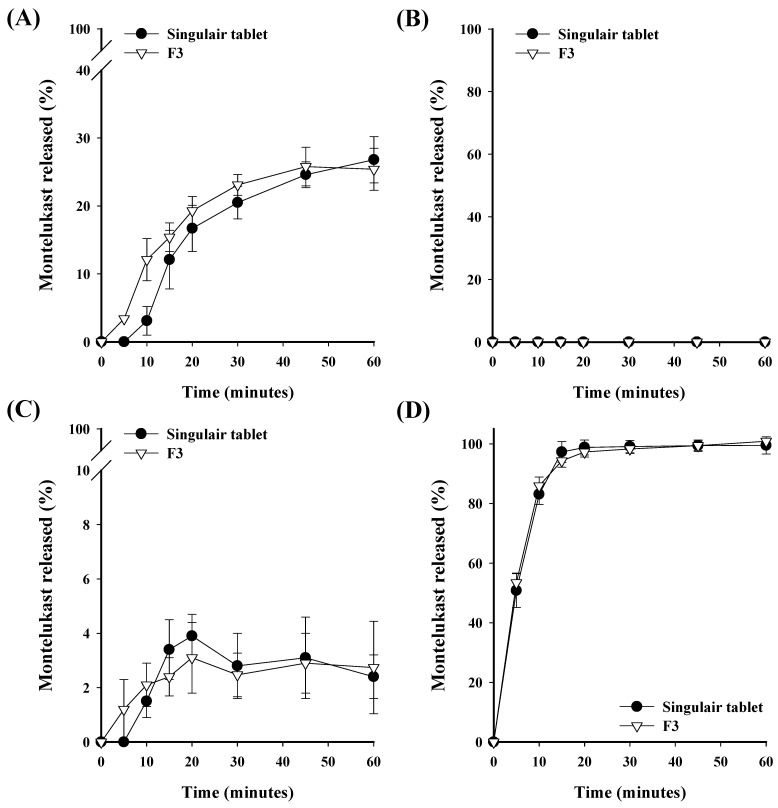
Comparison of the dissolution profiles of the montelukast–levocetirizine FDC monolayer tablet (F3 tablet) and montelukast commercial product (10 mg Singulair tablet) at (**A**) a pH of 1.2, (**B**) a pH of 4.0, (**C**) a pH of 6.8, and (**D**) in a 0.5% SLS. Each value represents the mean ± S.D. (n = 6).

**Figure 9 pharmaceutics-16-00963-f009:**
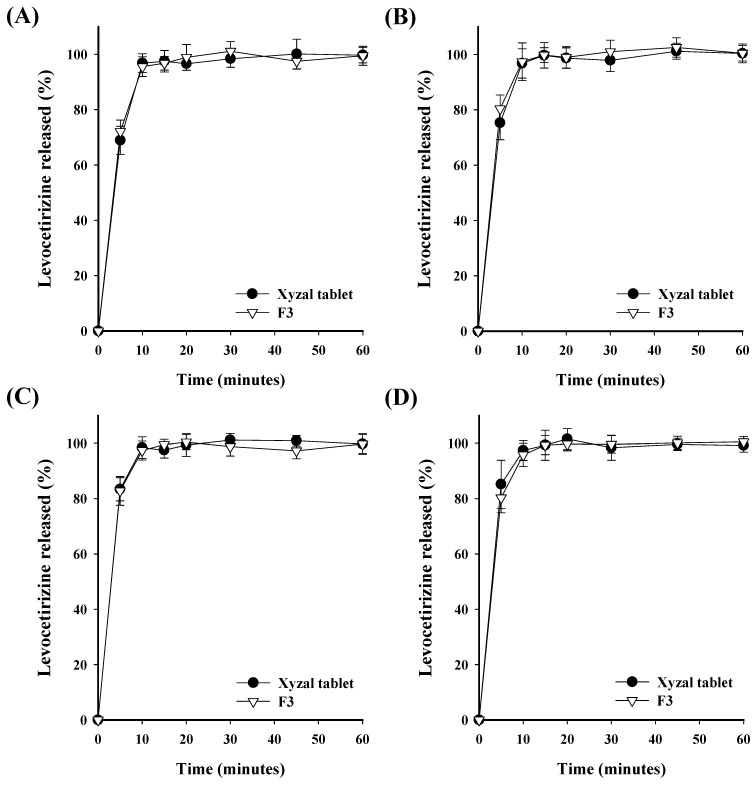
Comparison of the dissolution profiles of the montelukast–levocetirizine FDC monolayer tablet (F3 tablet) and levocetirizine commercial product (Xyzal 5 mg tablet) at (**A**) a pH of 1.2, (**B**) a pH of 4.0, (**C**) a pH of 6.8, and (**D**) in water. Each value represents the mean ± S.D. (n = 6).

**Figure 10 pharmaceutics-16-00963-f010:**
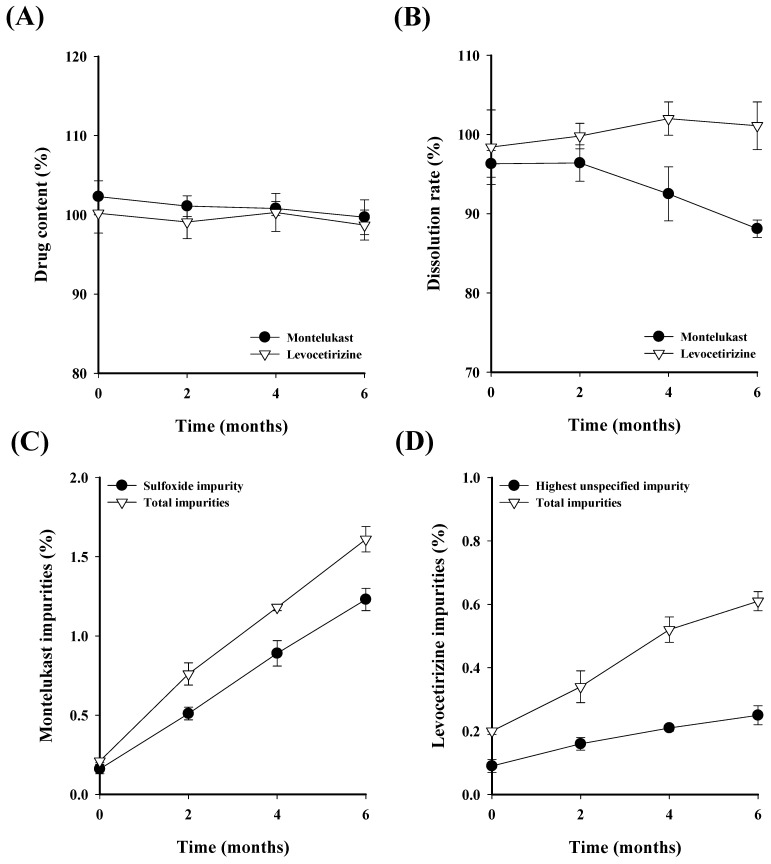
Stability test results over 6 months for the montelukast–levocetirizine FDC monolayer tablet (F3 tablet) regarding (**A**) drug content, (**B**) dissolution rate, (**C**) montelukast impurity content, and (**D**) levocetirizine impurity content.

**Table 1 pharmaceutics-16-00963-t001:** HPLC conditions for the impurity and dissolution rate analyses of montelukast and levocetirizine.

	Montelukast Impurity HPLC Method	Levocetirizine Impurity HPLC Method	Drug Content HPLC Method (Simultaneous Quantification)
Injection volume	10 μL	5 μL	20 μL
UVD wavelength	238 nm	230 nm	225 nm
Column	Zorbax SB-Phenyl, 4.6 mm × 250 mm, 5 μm	Symmetry Shield RP18, 4.6 mm × 250 mm, 5 μm	Hypersil GOLD-Phenyl, 3.0 mm × 100 mm, 5 μm
Column temperature	25 °C	30 °C	Room temperature
Flow rate	1.5 mL/min	1.0 mL/min	0.9 mL/min
Mobile phase	A. 0.1% (*v*/*v*) trifluoroacetic acid in water	A. Water/acetonitrile/10% trifluoroacetic acid = 690:300:10 (*v*/*v*/*v*)	A. 0.2% (*v*/*v*) trifluoroacetic acid in water
	B. 0.1% (*v*/*v*) trifluoroacetic acid in acetonitrile	B. Water/acetonitrile/10% trifluoroacetic acid = 290:700:10 (*v*/*v*/*v*)	B. 0.2% (*v*/*v*) trifluoroacetic acid in acetonitrile
Elution mode	Gradient ((min)/%A, %B) 0/60, 40 20/10, 90 30/10, 90 31/60, 40 35/60, 40	Gradient ((min)/%A, %B) 0/100, 0 2/100, 0 30/25, 75 40/100, 0 50/100, 0	Isocratic A:B = 60:40 (*v*/*v*) mixture

**Table 2 pharmaceutics-16-00963-t002:** Formulation compositions of the montelukast–levocetirizine FDC monolayer tablets.

Ingredients	Formulation (mg/Tablet)			
	F1	F2	F3	F4
Montelukast Wet Granulation				
Montelukast sodium	10.4	10.4	10.4	10.4
Mannitol 100SD	114.6	111.6	111.6	111.6
Avicel PH 101	60.0	60.0	60.0	60.0
Croscarmellose sodium	5.0	5.0	5.0	5.0
Sodium citrate	-	-	5.0	-
Montelukast Binder				
HPMC 2910 P645	3.0	3.0	3.0	3.0
(Water)	q.s	q.s	q.s	q.s
Levocetirizine Wet Granulation				
Levocetirizine dihydrochloride	5.0	5.0	5.0	5.0
Mannitol 100SD	-	100.0	100.0	100.0
Avicel PH101	-	44.0	44.0	44.0
Meglumine	-	-	-	5.0
Levocetirizine Binder				
HPMC 2910 P645	-	3.0	3.0	3.0
(Water)	-	q.s	q.s	q.s
Final Mixing				
Magnesium stearate	2.0	3.0	3.0	3.0
Total weight (mg)	200.0	345.0	350.0	350.0
Features of formulation	Montelukast granule + Levocetirizine post-blend	Montelukast granule + Levocetirizine granule	Sodium citrate added to montelukast granule	Meglumine added to levocetirizine granule

## Data Availability

Data available on request due to restrictions, e.g., privacy or ethical restrictions.

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
