# Peer review of "Enhanced Stability and Compatibility of Montelukast and Levocetirizine in a Fixed-Dose Combination Monolayer Tablet"

_pharmaceutics, 2024, doi:10.3390/pharmaceutics16070963_

Round 1

Reviewer 1 Report

Comments and Suggestions for Authors

pharmaceutics-3107250

Enhanced Stability and Compatibility of Montelukast and Levocetirizine in a Fixed-Dose Combination Monolayer Tablet

The manuscript described the development and study of the compatibility and stability of a monolayer tablet containing montelukast and levocetirizine. The authors presented sufficient data to support the conclusion. I have some suggestions below to improve the manuscript.

1. The Introduction should be revised to clarify the research gap as well as the novelty and contribution of this study. Please include more information about the FDC montelukast – levocetirizine.

2. Figure 8: The authors should change the Y-axes of (a), (b), and (c) to show the whole release profiles.

3. Figure 8: Regarding the low dissolution rate in pH 1.2, pH 4.0, and pH 6.8 of both F3 and Singulair, is it consistent with the literature of Singulair? Will it limit the BA of the drug? Any previous attempts to improve this? What is pH 4.0 mimicking?

4. Table 3 repeats data (dissolution rates) from previous Figures. They should be omitted.

Reviewer 2 Report

Comments and Suggestions for Authors

Line 99-103: can the Authors explain why a temperature of 60°C was used during drying? In lines 170-171, the Authors conclude that at this temperature there are unfavorable changes in the color of montelukast (see also Figure 1). The use of this temperature during the tablet mass preparation may have induced chemical degradation of montelukast.

Line 131-138: it is not clear how the test solutions were prepared. Are they solutions tested for compatibility or stability. Each substance was diluted differently, and how were the tablets diluted? No description of how the standard solutions were prepared.

Table 1: can the Authors provide a rationale for the choice of detection wavelength during the dissolution study?

Table 1: were the HPLC methods used developed by the Authors. If not, the Authors should provide references. If they were developed by the Authors, then were they validated/verified? Such information should be contained in the manuscript.

Line 144-145: it is not clear whether SLS was added to all dissolution medias or only to water. Were the same dissolution media used for both drug substances (in terms of SLS content)?

Line 227-229 - did the povidone used meet the requirements for pharmaceutical excipients? Can the authors explain what impurities and in what quantity are present in the povidone used? How can they affect the stability of the drug substance?

Figure 6: although the photo is not clear, one can see imperfections on the surface, especially in the score-line area, which suggests poor mechanical strength. Could the authors add the results of mechanical properties (e.g. hardness), LoD, etc. of the tablets at the beginning and after 4 weeks of storage?

Line 313-314: ”the characteristic elements S (sulfur) of montelukast sodium and Cl (chlorine) of levocetirizine dihydrochloride were analyzed“ - how do the authors distinguish between chlorine from montelukast molecules and chlorine from levocetirizine molecules?

Line 317-318: The spatial distribution of chlorine and sulfur looks identical, which is interesting because the description of the preparation of the tablets suggests that montelukast and levocetirizine should be concentrated in separate granules (two separate granulation processes). In addition, the both pictures show identical black areas, which do not contain elements characteristic of the two medicinal substances. Can the Authors comment on this?

Line 319-322: a description of the analytical method used to assess the uniformity of dosage units cannot be found in the manuscript.

Figure 8 and 9: why is the scale up to 120%? Especially with montelukast, this makes the curves poorly visible. “0.5% SLS” - does this mean an aqueous solution? “D.W.” - when first used, it should be explained what it means.

Table 3: did the authors consider the guidelines when calculating the factors f1 and f2? Especially the requirement “Not more than one mean value of > 85% dissolved for any of the formulations”. The table contains data for only one drug substance. What about the results for montelukast?

Comments on the Quality of English Language

The authors should take help in terms of improving the English language. There are many unfortunate sentences and statements in the text, e.g., in lines 123-124, 229,

Author Response

Please see the attachment. Additionally, I have appropriately revised the English according to your advice. I am also attaching the certificate. 

Round 2

Reviewer 1 Report

Comments and Suggestions for Authors

The manuscript was appropriately revised and can be accepted.

Reviewer 2 Report

Comments and Suggestions for Authors

I have no further comments